# Effect of X-Ray Tube Angulations and Digital Sensor Alignments on Profile Angle Distortion of CAD-CAM Abutments: A Pilot Radiographic Study

**DOI:** 10.3390/bioengineering12070772

**Published:** 2025-07-17

**Authors:** Chang-Hun Choi, Seungwon Back, Sunjai Kim

**Affiliations:** 1Graduate School, Yonsei University College of Dentistry, Seoul 03722, Republic of Korea; changhun81@gmail.com; 2Department of Prosthodontics, Gangnam Severance Dental Hospital, Yonsei University College of Dentistry, Seoul 06273, Republic of Korea; pote135283@naver.com

**Keywords:** periapical radiograph, accuracy, profile angle, X-ray tube angulation, digital sensor angulation

## Abstract

Purpose: This pilot study aimed to evaluate how deviations in X-ray tube head angulation and digital sensor alignment affect the radiographic measurement of the profile angle in CAD-CAM abutments. Materials and Methods: A mandibular model was used with five implant positions (central, buccal, and lingual offsets). Custom CAD-CAM abutments were designed with identical bucco-lingual direction contours and varying mesio-distal asymmetry for the corresponding implant positions. Periapical radiographs were acquired under controlled conditions by systematically varying vertical tube angulation, horizontal tube angulation, and horizontal sensor rotation from 0° to 20° in 5° increments for each parameter. Profile angles, interthread distances, and proximal overlaps were measured and compared with baseline STL data. Results: Profile angle measurements were significantly affected by both X-ray tube and sensor deviations. Horizontal tube angulation produced the greatest profile angle distortion, particularly in buccally positioned implants. Vertical x-ray tube angulations beyond 15° led to progressive underestimation of profile angles, while horizontal tube head rotation introduced asymmetric mesial–distal variation. Sensor rotation also caused marked interthread elongation, in some cases exceeding 100%, despite vertical projection being maintained. Profile angle deviations greater than 5° occurred in multiple conditions. Conclusions: X-ray tube angulation and sensor alignment influence the reliability of profile angle measurements. Radiographs with > 10% interthread elongation or crown overlap may be inaccurate and warrant re-acquisition. Special attention is needed when imaging buccally positioned implants.

## 1. Introduction

Periapical radiography remains a fundamental diagnostic tool for assessing marginal bone levels around dental implants, primarily due to its cost-effectiveness, accessibility, and ease of use in routine clinical settings [1,2]. Although periapical radiographs are limited in providing only the mesial and distal aspects of implants [3], they are regarded as sufficient for monitoring longitudinal changes in peri-implant bone levels [4,5], Moreover, periapical radiograph plays a critical role in the early detection of peri-implantitis and in evaluating progressive bone loss around implants [2]. However, periapical radiographs inherently involve projecting three-dimensional structures onto a two-dimensional plane, which introduces potential geometric distortion [6]. This limitation can result in the clinically significant underestimation of marginal bone loss [7,8,9].

Image distortion of periapical radiographs typically arises from two principal causes: improper angulation of the X-ray tube head and misalignment of the digital sensor (film) [1,10]. Even relatively modest vertical or horizontal deviations, particularly those exceeding 10 degrees from the ideal projection, have been shown to compromise the accuracy and reliability of radiographic measurements [11,12,13,14]. Suboptimal image quality and examiner variability may further influence measurement consistency [15,16].

Previous studies have predominantly investigated the effects of projection error on detecting marginal bone levels or peri-implant osseous defects [14]. However, with increasing interest in the role of restoration contour on peri-implant tissue health, attention has shifted toward more detailed morphological parameters, particularly the transmucosal configuration of implant restorations [17,18,19]. Katafuchi et al. reported that an emergence angle exceeding 30°, as measured on periapical radiographs, was significantly associated with a higher prevalence of peri-implantitis [20]. Furthermore, convex emergence profiles have been linked to a greater incidence of peri-implantitis than straight or concave profiles [20,21].

To better assess these morphological risks, Han et al. introduced the concept of the profile angle, a more refined metric designed to evaluate the three-dimensional geometry of CAD-CAM customized abutments [22]. Unlike prefabricated abutments, CAD-CAM abutments feature highly individualized, free-form configurations that pose challenges for accurate assessment using two-dimensional radiographs. The profile angle is measured by dividing the transmucosal area into three distance ranges from the implant–abutment junction (0–1 mm, 1–2 mm, and 2–3 mm) and calculating the angle between the implant’s long axis and the outer contour of the abutment within each distance range. Studies have shown that the profile angle is more predictive of marginal bone loss than the traditional emergence angle [22,23].

Despite its potential clinical relevance, measuring the profile angle from periapical radiographs presents methodological challenges. While the implant has a standardized geometry that may allow partial correction for angulation error, CAD-CAM abutments lack uniformity. CAD-CAM abutments are especially vulnerable to image distortion resulting from even minor tube head or sensor misalignment [24,25,26,27]. Since most current assessments of profile angle rely on two-dimensional radiography, it is essential to evaluate how changes in projection geometry affect the accuracy of these measurements, particularly in the context of CAD-CAM restorations where transmucosal configuration varies considerably.

No study has assessed the impact of vertical and horizontal tube head angulation or digital sensor rotation on profile angle measurements. Moreover, the influence of implant position relative to the center of the edentulous ridge, whether buccally or lingually offset, on the magnitude of radiographic distortion remains unclear. The current study aimed to investigate how variations in X-ray tube head angulation and sensor alignment affect the accuracy of profile angle measurements in CAD-CAM abutments. An additional comparison was performed on whether implant positioning within the residual ridge contributes to distortion in periapical radiographs.

## 2. Materials and Methods

A mandibular acrylic resin model (PRO2002-UL-HD-FEM-28; Nissin Dental, Kyoto, Japan) was used as a base model. The right and the second first molars were removed, and the empty areas were recontoured using modeling wax (Pinnacle Modeling Wax; Dentsply Sirona, Bensheim, Germany). The model was scanned using an intraoral scanner (TRIOS3; 3Shape, Copenhagen, Denmark), and the STL file was printed using a polyjet 3D printer (J5 DentaJet; Stratasys, Minnetonka, MN, USA) to fabricate a master model. To evaluate the effect of implant positioning on profile angle distortion, five implant locations were designated based on the bucco-lingual position of the screw access channel (SAC):

Group Cent: SAC at the occlusal fossa.Group B10: SAC 1.0 mm buccally off from a central position.Group B15: SAC 1.5 mm buccally off from a central position.Group L10: SAC 1.0 mm lingually off from a central position.Group L15: SAC 1.5 mm lingually off from a central position.

All implants were planned using digital implant planning software (Implant Studio1.7.83.0; 3Shape), and full-guided surgical templates were fabricated to ensure precise placement. A single master model was used with interchangeable implant removable dies, allowing consistent comparison of radiographic distortion across different implant positions.

Three CAD-CAM abutments were designed to represent identical profile angle characteristics from the buccal or lingual aspect. The profile angles were uniformly set to 15° in the 0–1 mm distance range (R1), 25° in the 1–2 mm range (R2), and 60° in the 2–3 mm range (R3) above the implant–abutment junction (IAJ). This standardization allowed for controlled comparisons by eliminating variability in the bucco-lingual emergence profile. In contrast, the abutment designs differed in the mesio-distal direction due to varying degrees of asymmetry based on the horizontal offset of the SAC:

S-Type (Symmetrical Abutment): designed for Group Cent, it featured a symmetrical profile in both the mesio-distal and bucco-lingual planes. In the mesio-distal view, the profile angles were 15°, 25°, and 40° in R1, R2, and R3, respectively (Figure 1A).

A10-Type (Asymmetrical 1.0 mm Offset): designed for Groups B10 and L10, presented asymmetry only in the mesio-distal plane. On the offset side (i.e., the direction of SAC displacement), the profile angles were 15°, 25°, and 60° in R1, R2, and R3. The opposite side featured a uniformly flattened profile angle of 7° across all three distance ranges (Figure 1B).

A15-Type (Asymmetrical 1.5 mm Offset): designed for Groups B15 and L15, exhibited the most pronounced asymmetry. The profile angles were 15°, 25°, 65° at the offset side, while the opposite side was designed with a fully vertical contour, featuring 0° in all three ranges (Figure 1C).

By keeping the bucco-lingual profile constant and only changing its shape when viewed from the mesio-distal direction (i.e., the proximal view), this design allowed us to investigate how different X-ray projection angles affect the measurement of profile angles depending on both the abutment contour and the implant′s position.

The right second premolar and second molar in the master model were prepared to receive full veneer ceramic crowns. The model was scanned using an intraoral scanner (TRIOS3; 3Shape), and full veneer restorations were designed using dental CAD software (CEREC inLab20; Dentsply Sirona). The crowns were then milled using a 3-axis milling machine (CEREC MC X; Dentsply Sirona) from lithium disilicate blocks (IPS e.max CAD; Ivoclar Vivadent, Schaan, Liechtenstein) and cemented using resin modified glass ionomer cement (RelyX Luting; 3M, Saint Paul, MN, USA) (Figure 2).

Each interchangeable implant die with its corresponding abutment was then sequentially positioned into the master model′s implant recipient site. Implant-supported crowns were fabricated following the same design and milling protocol used for the adjacent crowns. This process was repeated for all five implant dies, resulting in five implant-supported restorations, each corresponding to one of the experimental groups (Figure 3).

Radiographic imaging was performed with an intraoral X-ray device (ProXTM; Planmeca, Charlotte, NC, USA) and a CMOS digital sensor (RVG 6200; Carestream, Atlanta, GA, USA) under standardized conditions (70 kV, 1.12 mAs). Three experimental variables were tested: (1) Vertical tube head angulation: 0°, 5°, 10°, 15°, and 20° downward, with the digital sensor fixed parallel to the implant axis. (2) Horizontal digital sensor rotation: 0°, 5°, 10°, 15°, and 20° medial rotation, with the tube head fixed perpendicular to the implant. (3) Horizontal tube head angulation: 0°, 5°, 10°, 15°, and 20° mesial and distal rotations, with the digital sensor fixed.

Using CAD software (AutoCAD for MAC 2025 V.58.M.214; Autodesk Inc., San Francisco, CA, USA), interthread distances (between the second and third mesial threads), the amount of proximal overlap between adjacent restorations, and profile angles (mesial and distal aspects) were measured. Using CAD software (Meshmixer11.5.474; Autodesk Inc.), reference profile angles were obtained from the original STL abutment files. A calibrated examiner acquired each radiograph twice, and the mean value was used for analysis. As a pilot study with tightly controlled conditions and geometries, a single representative value per condition was analyzed descriptively. The focus was identifying distortion trends and absolute deviations rather than conducting inferential statistical comparisons.

## 3. Results

### 3.1. Interthread Distance Distortion by Vertical Tube Head Angulation

As the vertical projection angle of the X-ray tube head increased from 0° to 20°, the interthread distance consistently increased across all groups (Table 1). This distortion was most prominent in buccally positioned implants. At 20° vertical angulation, Groups B15 and B10 exhibited more than 15% increase in interthread distance, while Group L15 showed only a 78.5% increase. The Cent group (centrally positioned implants) demonstrated a moderate 11.2% increase under the same condition.

With the implant positioned at the center of the residual ridge (Group Cent), the amount of interthread distance distortion was expressed as a percentage, using the measurement at 0° vertical angulation of the radiation tube as the reference. Figure 4 displays a series of periapical radiographic images acquired with the vertical rotation angle of the radiation tube set at 0°, 5°, 10°, 15°, and 20°, from the leftmost to the rightmost image, respectively. Each image was captured while maintaining the specified angulation during exposure.

### 3.2. Interthread Distance Distortion by Horizontal Digital Sensor Rotation

Horizontal rotation of the digital sensor from 0° to 20° resulted in progressive increases in interthread distance across all groups (Table 2). Even at baseline (0°), Groups B15 and L15 demonstrated greater interthread distances than the Cent group, indicating the effect of implant position. As the rotation angle increased medially, the distortion became more pronounced. At 20°, the distortion in Group B15 was too severe to permit accurate measurement.

### 3.3. Crown Overlap by Distal Tube Head Angulation

Horizontal distal angulation of the tube head resulted in incremental overlap between the implant restoration and adjacent teeth (Figure 5). At 15°, approximately 5% overlap with the second premolar was observed in all groups, increasing to >10% at 20°. Overlap with the second molar consistently exceeded that of the premolar across all angulations (Table 3).

### 3.4. Crown Overlap by Mesial Tube Head Angulation

Mesial tube head angulation similarly led to progressive overlap, particularly between the implant crown and the second premolar. At 20°, crown overlap exceeded 10% in all groups except Cent, where image integrity remained acceptable (Table 4). As with distal rotation, overlap with the second molar occurred earlier and was more extensive than with the premolar.

### 3.5. Profile Angle Measurements

#### 3.5.1. Effect of Vertical Tube Head Angulation

With increasing vertical angulation (5°, 10°, 15°, and 20°), the measured profile angle decreased progressively. Relative to the 0° baseline, mean mesial deviations were 1.3°, 1.5°, 2.5°, and 3.6°, while distal deviations were 0.8°, 0.9°, 1.3°, and 2.1°, respectively. Profile angle discrepancies exceeding 5° were recorded in seven of 120 measurement conditions, with six occurring at 20° angulation and one at 15° (Table 5).

#### 3.5.2. Effect of Horizontal Digital Sensor Rotation

Sensor rotation also led to measurable distortion in profile angle measurements. When compared to the STL reference values (15°, 25°, 60°), the deviations ranged from −6° to +5°, with the maximum deviation (2.3°) observed in Group L15. Mean mesial deviations increased from 0.7° to 2.1°, and distal deviations from 0.8° to 2.1° across 5°, 10°, 15°, and 20° rotations (Figure 6 and Table 6).

#### 3.5.3. Effect of Horizontal Tube Head Angulation

Distal Angulation: Profile angle discrepancies exceeding 5° were identified in three mesial and six distal measurements out of one hundred and fifty total conditions. The maximum deviation was 10°, observed in Group B15 at 20° angulation. Mean mesial and distal deviations were 1.3° and 1.7°, respectively (Table 7).

Mesial Angulation: A total of four mesial and five distal measurements exhibited deviations greater than 5°. The maximum deviation (11°) occurred in Group B10 at 20° angulation. Mean mesial deviations across 5°, 10°, 15°, and 20° angulations were 0.9°, 1.6°, 1.4°, and 2.4°, respectively; distal deviations were 1.7°, 2.5°, 2.8°, and 4.3° (Table 8).

## 4. Discussion

Periapical radiographs are widely employed in implant dentistry to evaluate peri-implant marginal bone levels and, more recently, the transmucosal contours of CAD-CAM customized abutments [2,9,12]. However, these radiographs inherently project three-dimensional anatomical structures into two-dimensional images, making them susceptible to geometric distortion that may compromise the accuracy of radiographic assessments [6]. This study explored how deviations in X-ray tube head projection angles and digital sensor alignment affect the reliability of profile angle, a parameter increasingly utilized to evaluate the biological relationship between abutment contour and peri-implant tissue response.

Our findings demonstrated that vertical and horizontal projection errors can significantly distort profile angle measurements. Among the variables investigated, horizontal tube head angulation exerted the most substantial influence, introducing asymmetrical distortion that either exaggerated or underestimated mesial and distal profile angles depending on the direction of rotation. Vertical angulation beyond 15° led to a consistent underestimation of the measured profile angles.

Few studies have directly evaluated the impact of sensor malalignment on radiographic image distortion. Preus et al. reported that positioning a digital sensor at a 30° angle relative to the long axis of teeth in dry human mandibles led to significant elongation of object images, even when the X-ray tube was aligned correctly [28]. The current study showed that such rotation caused markedly greater distortion in interthread distances than vertical tube head deviations. Therefore, it is suggested that even minor sensor misalignment, often overlooked during paralleling techniques in clinical situations, can introduce significant image distortion. Thus, clinicians should exercise heightened caution in preventing sensor misalignment, as it may impact radiographic accuracy more than vertical angulation of the X-ray tube.

Another unique contribution of this study is the inclusion of implant positional offset within the residual ridge, which has not been addressed in prior literature. Even without projection angle deviations, 1.5 mm buccally off-center implants showed greater interthread distance elongation than lingually positioned implants. As projection angle deviation increased, distortion became more pronounced in buccally placed implants. At a vertical angulation of 20°, Group B15 exhibited nearly 40% interthread elongation, compared to only 7.9% in Group L15. Similarly, horizontal sensor rotation led to severe distortion in off-center groups, often exceeding acceptable thresholds or rendering measurement infeasible. This discrepancy is likely due to X-rays′ geometric divergence: when the implant is positioned closer to the X-ray source (e.g., buccally), the distortion increases because of the shorter source-to-object distance. Given that the distance from implant to sensor remains constant, these magnification effects become more pronounced. To mitigate this, increasing the distance between the X-ray tube head and the patient may help reduce beam divergence and minimize geometric distortion, particularly for buccally placed implants.

Based on the findings of this study, a set of clinically applicable guidelines can be proposed, radiographs exhibiting more than 10% interthread distance elongation or more than 10% crown overlap with adjacent teeth should be considered potentially inaccurate and retaken. These two thresholds can serve as practical indicators of vertical and horizontal projection error, respectively. This is especially relevant for buccally positioned implants, which appear more susceptible to distortion due to their proximity to the X-ray source. In such cases, clinicians may consider increasing the tube-to-object distance or using enhanced positioning protocols to maintain image fidelity.

Although this is a pilot study, it establishes critical trends, future studies with larger sample sizes and statistical analysis are warranted to validate these findings. A limitation of this pilot study is that the vertical angulation of the radiation tube was applied only in the positive direction. Previous studies have reported that when vertical angulation was increased in both positive and negative directions, greater linear distortion was observed in the negative direction at the same degree of rotation [14,29]. This aspect should be taken into consideration in the main study based on this pilot investigation. Further, in vivo studies, including patient-specific anatomical variations and comparisons with three-dimensional imaging modalities (e.g., CBCT with metal artifact reduction), would further strengthen the clinical relevance of profile angle assessments. Moreover, developing automated or AI-assisted tools for radiographic analysis may enhance standardization and reduce examiner-dependent variability, ultimately enabling a more robust evaluation of implant prosthesis design about peri-implant tissue health.

## Figures and Tables

**Figure 1 bioengineering-12-00772-f001:**
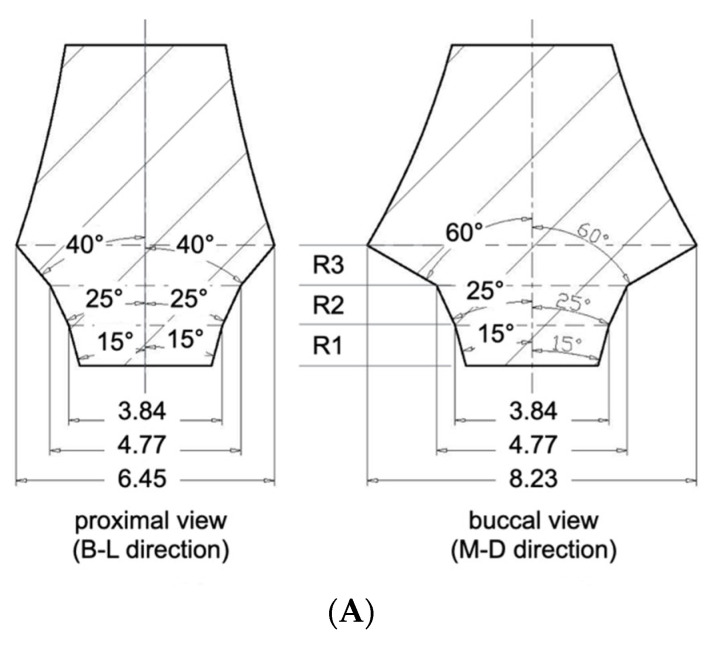
(**A**) S type CAD-CAM titanium customized abutment. R1; 0~1 distance range from implant–abutment junction, R2; 1~2 mm distance range from implant–abutment junction, R3; 2~3 mm distance range from implant–abutment junction. (**B**) A10 type CAD-CAM titanium-customized abutment. (**C**) A15 type CAD-CAM titanium-customized abutment.

**Figure 2 bioengineering-12-00772-f002:**
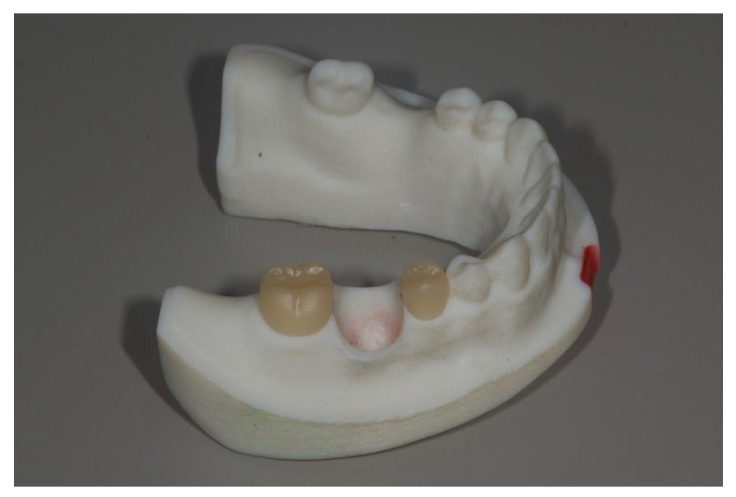
The right second premolar and the second molar received lithium disilicate (LS2) ceramic restorations. LS2 was chosen due to its translucent characteristics for proximal overlap measurements.

**Figure 3 bioengineering-12-00772-f003:**
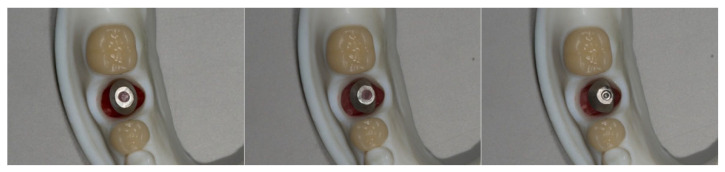
Three kinds of removable dies with an implant and its corresponding CAD-CAM abutment were placed into the recipient sites. From left to right, Group Cent, Group L10, Group L15.

**Figure 4 bioengineering-12-00772-f004:**
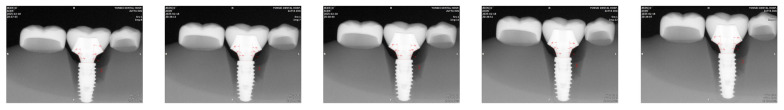
Periapical radiographic images of Group Cent taken from left to right with vertical angulations of 0°, 5°, 10°, 15°, and 20°. As the vertical angulation of the radiation tube increases, the interthread distance also increases.

**Figure 5 bioengineering-12-00772-f005:**
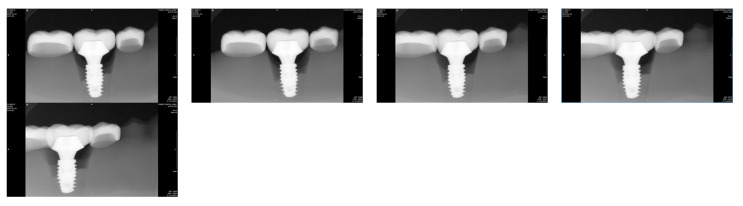
Periapical radiographic images taken from left to right with distal horizontal rotation of radiation tube at 0°, 5°, 10°, 15°, and 20° in Group B15.

**Figure 6 bioengineering-12-00772-f006:**
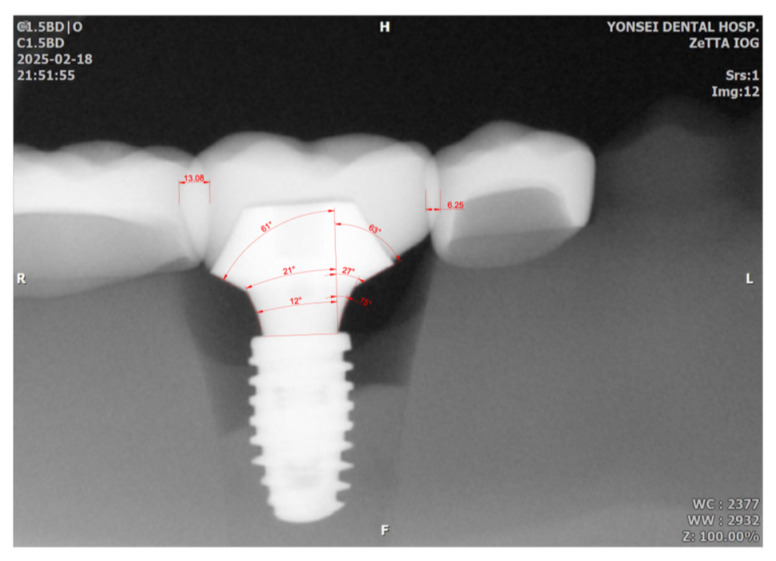
Measurements of mesial and distal profile angles and the amount of proximal overlap between adjacent restorations. Mesial and distal profile angles were measured across 3 distance ranges, resulting in 15°, 27°, 63° for the mesial aspect, and 12°, 21°, 61° for the distal aspect. The overlap values in the distal and mesial aspects were 13.08 and 6.25, respectively, in this image (the number represents units in the CAD program, not millimeters). The number was divided by the mesio-distal width of implant restoration (110.68 which is not shown in this picture) resulting in 11.8% and 5.6% for distal and mesial aspect of the implant restoration.

**Table 1 bioengineering-12-00772-t001:** The amount of elongation of interthread distance by vertical tube angulation (%).

	0	5	10	15	20
B15	−5.1	−3.2	0.5	2.3	15.9
B10	−2.2	1.3	3.8	5.9	17.2
Cent	0.0	0.5	2.4	7.8	11.2
L10	−1.6	0.2	0.8	2.3	9.9
L15	−1.7	0.7	1.9	2.5	8.5

**Table 2 bioengineering-12-00772-t002:** The amount of elongation of interthread distance by horizontal digital sensor angulation (%).

	0	5	10	15	20
B15	19.1	52.8	111.3	136.1	
B10	5.7	19.9	91.1	122.6	143.4
Cent	0.0	22.1	89.8	132.1	139.4
L10	15.1	55.5	94.9	128.6	136.4
L15	15.6	24.5	90.3	121.8	134.8

**Table 3 bioengineering-12-00772-t003:** The amount of overlap between the implant restoration and the adjacent restorations by distal horizontal rotation of radiation tube head (%).

	Second Premolar					Second Molar				
	0	5	10	15	20	0	5	10	15	20
B15	1.3	0.0	2.3	5.6	9.7	0.0	3.4	5.9	11.8	15.5
B10	1.3	0.0	1.7	4.9	9.6	0.0	1.5	3.9	7.7	13.0
Cent	0.8	0.0	2.8	5.2	9.3	0.0	2.3	4.9	9.7	13.9
L10	0.0	0.0	2.5	5.8	11.3	0.0	2.3	5.0	9.3	13.5
L15	1.7	0.0	2.8	4.8	10.0	0.0	2.8	5.9	10.0	15.4

**Table 4 bioengineering-12-00772-t004:** The amount of overlap between the implant restoration and the adjacent restorations by mesial horizontal rotation of radiation tube head (%).

	Second Premolar					Second Molar				
	0	5	10	15	20	0	5	10	15	20
B15	3.2	0.9	7.4	11.8		1.7	0.7	4.8	9.6	
B10	0.0	3.3	7.6	13.7		0.0	1.5	5.0	10.0	
Cent	0.0	3.6	7.1	11.4	13.8	1.0	0.0	3.9	7.5	11.7
L10	0.0	2.5	7.3	9.1		0.0	1.6	5.2	8.8	
L15	0.0	3.3	5.7	10.2		0.0	2.8	5.9	10.0	15.4

**Table 5 bioengineering-12-00772-t005:** Measured profile angle as the vertical rotation of radiation tube.

Vertical Rotation of Radiation Tube											
		0	5	10	15	20	0	5	10	15	20
			Mesial Profile Angle						Distal Profile Angle		
range 0~1	B15	15	13	15	12	12	12	14	15	15	13
	B10	14	13	11	12	14	14	13	13	14	14
	Cent	15	15	15	12	10	14	14	14	15	13
	L10	13	16	14	14	15	13	13	13	11	11
	L15	16	16	14	13	12	15	10	13	13	12
range 1~2	B15	26	23	21	20	18	23	25	24	223	23
	B10	25	24	23	23	22	23	23	23	23	21
	Cent	28	24	26	21	21	24	21	23	21	21
	L10	25	24	23	26	24	23	23	22	22	18
	L15	27	27	25	21	24	24	21	23	22	18
range 2~3	B15	61	61	59	58	55	59	60	59	58	56
	B10	60	60	59	59	58	61	60	60	58	56
	Cent	63	61	63	59	57	57	58	58	56	57
	L10	62	60	59	60	58	59	58	59	58	55
	L15	63	61	61	61	58	60	59	59	58	54

**Table 6 bioengineering-12-00772-t006:** Measured profile angles as the medial rotation of digital sensors.

Medial Rotation of Digital Sensor											
		0	5	10	15	20	0	5	10	15	20
			Mesial Profile Angle						Distal Profile Angle		
range 0~1	B15	14	14	14	15	16	15	14	14	14	16
	B10	14	14	14	15	16	14	13	16	14	18
	Cent	15	14	14	16	16	14	14	14	15	16
	L10	16	13	15	14	16	14	14	14	13	16
	L15	15	14	11	12	11	15	14	13	14	12
range 1~2	B15	23	24	23	27	24	24	23	22	24	25
	B10	25	25	26	26	24	24	25	23	24	25
	Cent	24	23	23	25	24	24	24	24	23	24
	L10	25	25	22	24	22	24	25	23	23	24
	L15	24	23	23	23	22	24	25	25	23	22
range 2~3	B15	61	61	60	63	65	61	61	63	64	66
	B10	60	61	62	63	65	60	59	62	63	66
	Cent	60	60	61	60	60	59	59	58	60	60
	L10	60	60	59	59	59	59	61	61	60	58
	L15	60	59	57	58	55	59	61	58	59	57

**Table 7 bioengineering-12-00772-t007:** Measured profile angles by horizontal distal rotation of radiation tube.

				Horizontal Distal Rotation of Radiation Tube							
		0	5	10	15	20	0	5	10	15	20
			Mesial Profile Angle						Distal Profile Angle		
range 0~1	B15	13	14	16	16	17	12	14	13	12	7
	B10	13	12	12	16	17	13	11	9	10	9
	Cent	12	12	14	16	19	14	13	14	13	11
	L10	15	13	14	16	17	15	12	14	12	11
	L15	12	13	13	15	14	15	15	14	13	13
range 1~2	B15	24	23	26	27	29	24	24	24	21	15
	B10	25	23	24	26	30	25	22	24	22	21
	Cent	23	24	24	27	28	23	23	20	22	25
	L10	25	25	25	24	28	25	23	23	23	22
	L15	25	23	25	24	25	25	25	23	24	23
range 2~3	B15	60	61	62	63	63	61	59	61	61	60
	B10	59	60	61	62	62	60	59	61	60	60
	Cent	61	60	59	60	60	60	61	62	61	60
	L10	61	60	59	60	60	59	60	60	59	59
	L15	59	61	59	60	60	61	60	60	61	60

**Table 8 bioengineering-12-00772-t008:** Measured profile angles by horizontal mesial rotation of radiation tube.

				Horizontal Mesial Rotation of Radiation Tube							
		0	5	10	15	20	0	5	10	15	20
			Mesial Profile Angle						Distal Profile Angle		
range 0~1	B15	13	11	13	11	10	11	15	16	16	19
	B10	14	14	11	13	11	14	16	17	18	25
	Cent	14	13	13	13	13	15	14	15	17	20
	L10	14	13	11	13	12	13	16	17	18	17
	L15	13	13	11	11	12	15	17	18	18	18
range 1~2	B15	24	22	21	21	18	23	26	26	27	29
	B10	22	22	20	21	16	25	26	28	28	31
	Cent	22	25	25	24	23	24	24	27	29	31
	L10	23	23	24	23	25	24	27	27	26	25
	L15	23	23	22	21	24	24	25	27	26	27
range 2~3	B15	61	60	59	60	57	60	61	63	63	65
	B10	59	61	61	62	59	62	61	63	63	64
	Cent	60	60	60	60	62	61	59	60	60	61
	L10	61	60	61	62	63	61	60	62	59	60
	L15	60	61	59	61	62	60	60	61	60	62

## Data Availability

The original contributions presented in this study are included in the article. Further inquiries can be directed to the corresponding author.

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
