# Peer review of "Effect of X-Ray Tube Angulations and Digital Sensor Alignments on Profile Angle Distortion of CAD-CAM Abutments: A Pilot Radiographic Study"

_bioengineering, 2025, doi:10.3390/bioengineering12070772_

Round 1

Reviewer 1 Report

Comments and Suggestions for Authors

It is an interesting study on a clinical topic. There are however issues that must be corrected before acceptance

- The authors have used only   Vertical tube head angulation: 0°, 5°, 10°, 15°, and 20° downward. Why not include also 5°, 10°, 15° upwards?  The angle of the x-ray tube to the axis of the implant definitely affects the projected space between the implant neck and the abutments. There is a publication focused on it that is not mentioned neither in the Introduction nor in the Discussion. 

Radiographical Evaluation of the Gap at theImplant-Abutment Interface.  J Esthet Restor Dent 22:235–251, 2010, DOI 10.1111/j.1708-8240.2010.00345.x
 In this paper the effect of the tube angle has been clearly shown. 

-Apart from thr above mentioned more photos should accompany the text to show the effect of the angulation both in horizontal and vertical dimensions. 

Author Response

COMMENT 1:  The authors have used only   Vertical tube head angulation: 0°, 5°, 10°, 15°, and 20° downward. Why not include also 5°, 10°, 15° upwards?  The angle of the x-ray tube to the axis of the implant definitely affects the projected space between the implant neck and the abutments. There is a publication focused on it that is not mentioned neither in the Introduction nor in the Discussion. 

Radiographical Evaluation of the Gap at theImplant-Abutment Interface.  J Esthet Restor Dent 22:235–251, 2010, DOI 10.1111/j.1708-8240.2010.00345.x
 In this paper the effect of the tube angle has been clearly shown. 

Answer 1: Thank you for your insightful comment. As you correctly pointed out, the study by Papavassiliou et al. indeed concludes that in external hex implants, “In angulation with inclination to the implant (− degrees), the gap disappeared faster compared with inclination to the prosthetic abutment (+ degrees).” This observation is further supported by the data in the results section of their article, where angulations in the negative direction consistently showed greater dimensional changes (Dist.y %) compared to equivalent positive angles.

Similarly, in the study by Rugani et al. cited in our references, when the horizontal angulation was fixed at 0° and vertical angulation was varied at 0, ±5, ±10, and ±15°, the linear distortion at ±5° was approximately 9% in both directions. However, at ±10°, distortion increased to 15% for +10° and 19% for −10°, and at ±15°, distortion was 18% for +15° and 20% for −15°, indicating a consistently greater distortion in the negative direction.

At the time of designing the pilot study, we assumed that the direction of vertical rotation (positive vs. negative) of the X-ray tube would not significantly affect the magnitude of distortion, and therefore assessed only the positive angulation. However, based on your valuable suggestion and the supporting literature, we agree that vertical angulation in the negative direction should also be evaluated.

Accordingly, we have added this point to the Discussion section as a limitation of the current pilot study, and we will incorporate symmetrical angulations (both positive and negative directions) in the design of the main experiment moving forward.

Comment 2: Apart from thr above mentioned more photos should accompany the text to show the effect of the angulation both in horizontal and vertical dimensions. 

Answer 2: As per your valuable suggestion, we have added two additional figures to illustrate the radiographic effects more clearly. Figure 4 demonstrates the changes in interthread distance according to increasing vertical angulation. This figure also includes the corresponding changes in profile angle resulting from variations in vertical angulation.

Additionally, Figure 5 has been included to show the overlap between the implant restoration and the adjacent restorations as horizontal tube rotation increases.

We appreciate your insightful feedback, which helped enhance the clarity and completeness of our visual data presentation.

Reviewer 2 Report

Comments and Suggestions for Authors

This is a well-organized pilot study that investigates the influence of radiographic geometry on the accuracy of profile angle measurements in CAD-CAM abutments. However, as acknowledged by the authors, the pilot nature and limited sample size warrant cautious interpretation of the findings. Furthermore, lack of statistical analysis limits its use and the strength of the conclusions. 

The paper makes  recommendations for acceptable thresholds for interthread elongation (>10%) and crown overlap. However, they appear arbitrary. Further details should be included to explain whether they are based on previous literature or expert consensus, or based on observed trends in this study.

The buccally positioned implants showed greater distortion due to beam divergence is an important finding. A direct statement about how this could influence clinical image acquisition or abutment design considerations can be beneficial. 

Author Response

Comment 1: The paper makes  recommendations for acceptable thresholds for interthread elongation (>10%) and crown overlap. However, they appear arbitrary. Further details should be included to explain whether they are based on previous literature or expert consensus, or based on observed trends in this study.

Answer 1: You are absolutely right. As a pilot study, this research has a major limitation in that the evaluation was based on a single radiographic image per condition, without multiple acquisitions under identical settings. As such, no statistical analysis was performed, which significantly limits the generalizability of the findings.

We acknowledge this limitation, and based on the findings of this preliminary study, we are preparing the main study to include a larger number of images under controlled conditions to allow for appropriate statistical evaluation and more meaningful conclusions.

Regarding your comment about the 10% threshold, we fully agree. In this pilot study, when the vertical angulation exceeded 20°, we observed a sudden increase in interthread distance distortion exceeding 10%. At this level of distortion, errors in profile angle measurements also began to exceed 5°, which we believe can critically affect the accuracy of the radiographic analysis. For this reason, we suggested that images with distortion greater than 10% (typically resulting from radiation tube angulation exceeding 20°) should be retaken.

However, as mentioned earlier, this recommendation was not statistically validated due to the limited design of the pilot study. In order to acknowledge this clearly, we have added the following sentence in the Discussion section to further emphasize this limitation:

“However, this finding should be further justified in the main study, which will be based on this pilot experiment and will involve the analysis of a larger number of images with appropriate statistical evaluation.”

We thank the reviewer for highlighting this critical point, which has helped us to better clarify the limitations and future directions of our study.

Comment 2: The buccally positioned implants showed greater distortion due to beam divergence is an important finding. A direct statement about how this could influence clinical image acquisition or abutment design considerations can be beneficial. 

Answer 2: Thank you for your valuable comment. As you pointed out, this is an important consideration. To address this issue, we have added the following sentence to the Discussion section as a possible solution:

“To mitigate this, increasing the distance between the X-ray tube head and the patient may help reduce beam divergence and minimize geometric distortion, particularly for buccally placed implants.”

We appreciate your thoughtful feedback, which helped improve the clarity and clinical relevance of our discussion.

Reviewer 3 Report

Comments and Suggestions for Authors

I believe this is a clinical subject of interest. However, despite being an experimental study, there was a lack of scientific rigor in the radiographic exposures, such as: use of a radiographic positioner, explanation of how the radiographic exposure angles were measured. Thus, the conditions and geometry were not well controlled. Of course, this greatly interferes with the results found.

Comments on the Quality of English Language

Not comment

Author Response

Comment: I believe this is a clinical subject of interest. However, despite being an experimental study, there was a lack of scientific rigor in the radiographic exposures, such as: use of a radiographic positioner, explanation of how the radiographic exposure angles were measured. Thus, the conditions and geometry were not well controlled. Of course, this greatly interferes with the results found.

Answer: Thank you very much for your thoughtful and constructive comment. We fully agree with your observations. Although this study was designed as a pilot, we acknowledge that achieving more accurate and reliable results requires the use of a more precise supporting apparatus, along with a clear and detailed description of the measurement setup.

In this study, we used the Rinn device along with a protractor to control angulation. However, the custom apparatus connecting these components—comprising a ruler, protractor, and hardboard—was not included in the manuscript due to its rudimentary appearance and lack of visual clarity. While this may seem like an insufficient excuse, we genuinely recognize the need for transparency and technical rigor.

Based on the foundational insights gained from this pilot study, we are currently preparing the main study with a more refined device and a larger number of radiographic images to ensure statistically meaningful and clinically relevant outcomes.

We sincerely thank you once again for taking the time to review our work and for providing invaluable guidance to improve the quality of our research.

Round 2

Reviewer 1 Report

Comments and Suggestions for Authors

The auhtors have made all suggested changes. I suggest acceptance

Reviewer 2 Report

Comments and Suggestions for Authors

No further comments.

Reviewer 3 Report

Comments and Suggestions for Authors

I'd like to see the main paper, when you describe more scientific rigor in the radiographic exposures, such as: use of a radiographic positioner, explanation of how the radiographic exposure angles were measured.